# Immuno-Microbial Signature of Vaccine-Induced Immunity against SARS-CoV-2

**DOI:** 10.3390/vaccines12060637

**Published:** 2024-06-07

**Authors:** Lesley Umeda, Amada Torres, Braden P. Kunihiro, Noelle C. Rubas, Riley K. Wells, Krit Phankitnirundorn, Rafael Peres, Ruben Juarez, Alika K. Maunakea

**Affiliations:** 1Department of Molecular Biosciences and Bioengineering, University of Hawaii, Honolulu, HI 96822, USA; umedal@hawaii.edu (L.U.); bradenku@hawaii.edu (B.P.K.); nrubas@hawaii.edu (N.C.R.); rkwells@hawaii.edu (R.K.W.); 2Department of Anatomy, Biochemistry, and Physiology, John A. Burns School of Medicine, University of Hawaii, Honolulu, HI 96822, USA; torres91@hawaii.edu (A.T.); kritphan@hawaii.edu (K.P.); peres@hawaii.edu (R.P.); 3Department of Economics and UHERO, University of Hawaii, Honolulu, HI 96822, USA; rubenj@hawaii.edu; 4Hawaii Integrated Analytics, LLC, Honolulu, HI 96822, USA

**Keywords:** HMGB-1, inflammation, SARS-CoV-2, gut microbiome, vaccine

## Abstract

Although vaccines address critical public health needs, inter-individual differences in responses are not always considered in their development. Understanding the underlying basis for these differences is needed to optimize vaccine effectiveness and ultimately improve disease control. In this pilot study, pre- and post-antiviral immunological and gut microbiota features were characterized to examine inter-individual differences in SARS-CoV-2 mRNA vaccine response. Blood and stool samples were collected before administration of the vaccine and at 2-to-4-week intervals after the first dose. A cohort of 14 adults was separated *post hoc* into two groups based on neutralizing antibody levels (high [HN] or low [LN]) at 10 weeks following vaccination. Bivariate correlation analysis was performed to examine associations between gut microbiota, inflammation, and neutralization capacity at that timepoint. These analyses revealed significant differences in gut microbiome composition and inflammation states pre-vaccination, which predicted later viral neutralization capacity, with certain bacterial taxa, such as those in the genus *Prevotella*, found at higher abundance in the LN vs HN group that were also negatively correlated with a panel of inflammatory factors such as IL-17, yet positively correlated with plasma levels of the high mobility group box 1 (HMGB-1) protein at pre-vaccination. In particular, we observed a significant inverse relationship (Pearson = −0.54, *p* = 0.03) between HMGB-1 pre-vaccination and neutralization capacity at 10 weeks post-vaccination. Consistent with known roles as mediators of inflammation, our results altogether implicate HMGB-1 and related gut microbial signatures as potential biomarkers in predicting SARS-CoV-2 mRNA vaccine effectiveness measured by the production of viral neutralization antibodies.

## 1. Introduction

Severe acute respiratory syndrome coronavirus 2 (SARS-CoV-2) is a novel coronavirus that causes the highly infectious coronavirus disease 2019 (COVID-19). COVID-19 was initially documented in December 2019 and its subsequent spread resulted in a global pandemic in 2020. The primary mode of transmission is through inhalation and direct contact with respiratory secretions and droplets. Those infected can suffer from flu-like symptoms including coughing, respiratory failure, acute respiratory distress syndrome and even death [1]. These conditions caused hospitals and medical centers to be overwhelmed around the world. To address the spread of the disease, vaccine development began early in the pandemic, with the mRNA vaccines, Moderna Spike Vax, and the Pfizer-BioNTech COMIRNATY vaccines being approved for emergency distribution in late 2020 in the United States of America [2]. These vaccines were administered in a series of two initial doses, with several subsequent doses given as boosters.

The effectiveness of vaccines, herein defined as the capacity to produce antibodies capable of binding to the targeted antigen and enabling viral neutralization, between individuals can vary due to a variety of factors, from environmental to genetic variation, with the immune system and the intestinal microbiota playing a major role as determinants in vaccine response [3,4]. Traditionally, vaccine development has focused on the average vaccine response and effectiveness at the population-level. This approach may lead certain individuals, especially those that are immunocompromised, more vulnerable than others to infection and higher severity of disease despite vaccination. Determining vaccine effectiveness and identifying potential modifiers of vaccine-induced immunity at the individual-level is important to improve outcomes and reduce potential side effects. Identifying underlying factors that contribute to inter-individual variability in vaccine response is requisite to predicting vaccine outcomes and potentially tailoring vaccines with adjuvants that can overcome poor effectiveness. A crucial step in infectious disease prevention and mitigation will involve understanding and personalizing vaccines based on individual immunomodulating characteristics.

As an important modulator of the immune system, the gut microbiome may play a significant role in influencing vaccine effectiveness [4,5,6]. An imbalance in the gut microbiome resulting in increased inflammation may contribute to a decrease in the effectiveness of a vaccine through damage to the mucosal layer or general cellular damage [4]. Meanwhile, short-chain fatty acids (SCFAs) produced by intestinal microbiota through anaerobic fermentation have been shown to influence energy metabolism to support antibody production in B-cells, modulate plasma B-cell differentiation, and influence active immune system responses [7,8]. Butyrate is a prominent SCFA that increases mucosal immunity, improves intestinal barrier integrity, and has been observed to have anti-inflammatory properties. Butyrate production is also associated with health outcomes such as reduced obesity and improved brain function [5,6,8].

Associated with gut dysbiosis and poor diets [9], chronic low-grade inflammation is another factor that is known to modulate vaccine response. If left uncontrolled, chronic inflammation is also associated with the development of more severe outcomes from viral infection [10]. An increased presence of pro-inflammatory innate immune cells correlated with a decreased response to the hepatitis B virus vaccination in aged individuals [11]. Chronic inflammation often found in obese individuals is also associated with a lower capacity to neutralize a viral infection [12]. With the SARS-CoV-2 vaccine, there are reported cases of inflammation that may be exacerbated by a pre-existing inflammation status vaccination. Understanding the pre-vaccination inflammatory status and its relationship to the gut microbiome will help identify biomarkers that may predict an individual’s vaccine response and enable personalized medicine. 

High Mobility Group Box 1 protein (HMGB-1) is a non-histone DNA-binding protein that facilitates DNA repair and replication in the nucleus, and in the cytoplasm serves as a signaling molecule that functions as a trigger for inflammation by attaching to receptors, attracting inflammatory agents, cytokines, and macrophages [13]. HMGB-1 is known to act as a chemotactic cytokine and conjugates with other receptors involved in inflammation including the potent induction of the IL-1β pro-inflammatory response. Given these multimodal properties, we investigated inter-individual variability of HMGB-1 and associations with anti-SARS-CoV-2 antibody production induced by vaccination. Specifically, in this longitudinal observational pilot study, we report changes in the immune response and gut microbiome of 14 individuals receiving a two-dose SARS-CoV-2 mRNA vaccine from either Pfizer-BioNTech or Moderna over a 10-week period. Together with shifts in the gut microbiome, our results implicate HMGB-1 as a potential biomarker for predicting vaccine effectiveness measured by in vitro neutralization assays.

## 2. Materials and Methods

### 2.1. Participant Recruitment and Consent

Participants were enrolled in a 10-week longitudinal observational pilot study. The study inclusion criteria required that participants had not yet received any SARS-CoV-2 vaccine and expressed willingness to receive it within the subsequent two weeks from study entry. Additionally, participants were required to adhere to the sample collection schedule, remain available for sample collection, and provide consent for participation in the study. Individuals exhibiting signs of infectious disease, or four or more symptoms of SARS-CoV-2, were ineligible for participation. All samples for this study were collected from January to April 2021 in Hawaii following procedures approved by the University of Hawaii Human Studies Institutional Review Board. The study protocol, identified as Protocol ID 2019-00376 and accepted on 1 May 2021, was overseen by the University of Hawaii Office of Research Compliance. All operations strictly adhered to the guidelines specified in the approved protocol. All experiments were performed in accordance with relevant guidelines and regulations.

### 2.2. Sample and Data Collection

Participants self-reported answers to questions on demographics, age, sex, ethnicity, period of residency in Hawaii, brand of mRNA vaccine received, and date of first and second doses. Weight and height were collected at an in-person exam at the University of Hawaii laboratory site and used to calculate the Body Mass Index (BMI). After the first and second doses, participants were asked to report possible side effects, such as headache, fatigue, muscle pain, nausea, vomiting, fever, shaking chills, rhinorrhea (runny nose), joint pain, or any other symptoms of discomfort (sore throat and cough).

Venous blood samples were collected before and after the administered inoculum for the period of 10 weeks in intervals of two weeks. Samples were processed within 24 h of collection using Sepmate tubes (Stemcell, Vancouver, BC, Canada) to isolate plasma. Stool sample self-collection kits were distributed to participants prior to vaccination and at four-week intervals until the conclusion of the 10-week study. Instructions for proper stool sample collection and storage were provided both verbally and in print. Samples were collected and stored at −20 °C by a community facilitator within one week after kit distribution. DNA and RNA were isolated using MagMAX Microbiome Ultra Nucleic Acid Isolation kits (Thermo Fisher Scientific, Inc., Waltham, MA, USA) and stored at −80 °C. An overview of the workflow and timing for blood and stool sample processing and analyses are illustrated in Figure 1.

### 2.3. Antiviral Response: Anti-SARS-CoV-2 Neutralization Titer

Plasma samples were subjected to the SARS-CoV-2 surrogate virus neutralization test (GenScript, Piscataway, NJ, USA), which identifies total immunodominant neutralizing antibodies directed against the viral spike protein receptor-binding domain (RBD). The test was conducted at a dilution of 10×, following the GenScript protocol. The absorbance of the sample exhibits an inverse relationship with the titer of anti-SARS-CoV-2 neutralizing antibodies. Neutralization capacity was quantified as a percentage based on standard curve following the manufacturer’s instructions.

The Human Coronavirus Ig Total 11-Plex ProcartaPlex Panel (Thermo Fisher Scientific, Inc., Waltham, MA, USA) was used to measure four anti-SARS-CoV-2 antibodies: spike trimer protein (S), S1 subunit, RBD, and nucleocapsid. Fluorescent signals were obtained and data processing was carried out as we previously described [14]. Due to potential cross-reactions with all immunoglobulin proteins (IgG, IgM, and IgA) reported by the Luminex plate, additional validation and characterization was performed with the SeroFlash SARS-CoV-2 IgG/IgM ELISA Fast Kit (EpigenTek Inc., Farmingdale, NY, USA) to simultaneously detect both IgG and IgM antibodies using a lateral flow colloidal gold-based immunoassay to quantify the antibodies against the SARS-CoV-2 spike protein in the plasma samples. All enzyme-linked immuno-sorbent assays (ELISA) were read on a SpectraMax ABS/ABS Plus Microplate Reader (Molecular Devices, Sunnyvale, CA, USA).

### 2.4. Immune Response Profiling

A multiplex, bead-based ProcartaPlex™ Human Immune Monitoring Panel, Luminex, 65 plex (Thermo Fisher, Waltham, MA, USA) was used according to manufacturer instructions to assess plasma cytokine concentrations. Cytokine levels were measured before vaccination, and at four weeks and six weeks following the vaccination. Fluorescent signals were analyzed using the Luminex 200 instrument (R&D Systems, Inc., Minneapolis, NE, USA) and data were processed using Bio-Plex Manager™ software (RRID:SCR 014330, Bio-Rad Laboratories, Inc., Hercules, CA, USA). Qiagen’s ingenuity pathway analysis software (IPA v01-2201, Qiagen Inc., Valencia, CA, USA) was used to determine enriched pathways.

Additional plasma levels of immunological biomarkers of interest were quantified using ELISA. For the C-reactive protein (CRP), the Thermo Fisher Instant ELISA assays were performed according to manufacturer instructions using BMS288INST and BMS2007INST kits. Samples above the standard curve were further diluted and assayed for a second time. HMGB-1 and nitric oxide (NO) were measured using a Thermo Fisher ELISA Kit number EEL047 and EH169RB, respectively, without sample dilution.

### 2.5. Gut Microbiota Evaluation

Quality and concentration of nucleic acids extracted from stool samples were assessed using the NanoDrop Microvolume Spectrophotometer kits (Thermo Fisher Scientific Inc., Waltham, MA, USA). The 16s hypervariable regions V2-4 and V6-9 were amplified with polymerase chain reaction (PCR) on 40 ng of DNA extracted from stool samples using the Ion Torrent 16S^TM^ Metagenomics Kit (Thermo Fisher). Amplicon products were pooled from each primer set and purified using magnetic beads. A Qubit dsDNA BR Assay was used to quantify the PCR products with 150 ng of pooled amplicons (Ion Plus Fragment Library Kit; Thermo Fisher Scientific, Waltham, MA, USA) and barcoded using Ion Xpress Barcode Adapters. DNA libraries were quantified using qPCR and diluted uniformly to 80 pmol before being loaded onto Ion 530™ chips and sequenced using the Ion S5 Next-Generation Sequencing System.

16s Metagenomics Kit analysis was performed using Ion Reporter™ Software v5.18.4.0 (Thermo Fisher Scientific Inc., Waltham, MA, USA). Sequenced fragments or “reads” were mapped to reference databases Greengenes v13.5 and MicroSEQ ID v3.0, and gut microbiome profiles were compiled using the Curated MicroSEQ(R) 16S Reference Library v2013.1. Raw abundance values were subsampled at 10,000 reads per sample, excluding samples with fewer than 10,000 total reads, to control for inequivalent read numbers across samples. The rrarefy function of the Vegan R package was performed at the species-level operational taxonomic unit (OTU) table [15]. Family-level OTU tables were preferred due to large amounts of unclassified upstream classifications when using genus and species-level OTU tables. The family-level OTU data was compared to the NCBI database via the classification function of the taxize R package to determine upstream taxonomic ranks [16]. Genus and species-level OTU tables were joined onto the family-level OTU table to form a comprehensive taxonomic classification. Subsampled species-level reads were converted to per sample relative abundance values via the “transform” function of the microbiome R package [17]. Significant differences in gut microbiome taxa were determined through the Wilcoxon signed rank test and represented in a heat map using z-transformed abundance.

### 2.6. Statistical Data Analysis

The study cohort was divided *post hoc* based on SARS-CoV-2 neutralization ability at the tenth week, using the 50th percentile as a cut-off threshold. This resulted in two groups: low neutralization (LN) and high neutralization (HN). To examine differences in variables of interest between the LN and HN groups, unpaired *t*-tests with Welch’s correction were performed. Neutralization capacity and antibody production were evaluated at two-week intervals using non-parametric Kruskal-Wallis tests followed by *post hoc* Dunn’s multiple comparison tests on group means. Using Pearson correlation coefficient analysis, plasma concentrations of HMGB-1 were correlated with the neutralization antibody capacity at 6, 8, and 10 weeks after vaccination, and with NO production at pre-vaccination and at week 10 post-vaccination. The Spearman’s rank correlation coefficients between the immunological biomarkers of interest and differentially abundant microbial species were calculated using the corrplot Bioconductor package (v0.92). Graphing and statistical analyses were performed using Prism (v9.0c, GraphPad Software, La Jolla, CA, USA). For all analyses, the significance level was established at *p* < 0.05.

## 3. Results

In this longitudinal observational pilot study, a cohort of fourteen healthy individuals participated in a 10-week protocol to explore the potential impact of individual immunological status and intestinal microbiota composition on the effectiveness of two mRNA SARS-CoV-2 vaccines, Pfizer (BNT162b2, Pfizer-BioNTech COVID-19) or Moderna (COVID-19 Vaccine SPIKEVAX, monovalent or Bivalent). The cohort consisted of eight men and six women, with an average age of 32 years (ranging from 22 to 52 years) at study entry. About sixty percent were of Asian and Pacific indigenous origin, including Native Hawaiians and Pacific Islanders (NHPI), and East Asians and Southeast Asians. The non-Asian group included people of Hispanic or Caucasian ethnic origin. The average residence time of participants in Hawaii was 14 years, ranging from those who had spent most of their lives in the state to those who had lived in Hawaii for a minimum of three years prior to enrolling in the study. According to the NIH BMI classification, the group was in the normal and overweight range with a mean of 25.3 kg/m^2^. None of the participants were obese. All participants took two doses of the same brand of the respective mRNA vaccine according to the inoculation protocols accepted on the date of the vaccination. Adverse symptoms experienced by participants during the study included fatigue (86%), headache (57%), nausea (36%), chills (29%), fever (15%), body aches (7%), runny nose (7%), and cough (7%) (Table 1 and Appendix A).

To explore differences in eventual vaccine-mediated viral neutralization, the cohort was divided into groups based on SARS-CoV-2 neutralization ability at the tenth week, using the 50th percentile as a cut-off threshold determined *post hoc*, resulting in LN or HN groups. To evaluate potential immunization bias in our data, we initially assessed baseline neutralization ability pre-vaccination, which demonstrated an average neutralization capacity of 10% (±7.0) with no significant differences in inhibition between the two neutralization groups. There were also no significant differences between the two groups regarding vaccine brands and time between the vaccine doses. These groups were well balanced, with no statistical differences observed between them for variables listed in Table 1.

### 3.1. SARS-CoV-2 Neutralization and Antibody Production

The responses to vaccine doses were evaluated longitudinally at two-week intervals. SARS-CoV-2 neutralization capacity was determined using a surrogate virus ELISA, which challenges the RBD protein, and by measuring levels of circulating SARS-CoV-2 neutralizing antibodies using a Human Coronavirus Ig Total 11-Plex panel on Luminex technology. Immune response profiling for the entire cohort revealed increases in SARS-CoV-2 neutralizing antibodies and antibody production against various regions of the spike protein, including the trimer, S1 protein, and RBD antigens, observed at all time points following the initial mRNA vaccination (Figure 2 and Appendix A). Immune response was significantly elevated at two timepoints, particularly from baseline to week 2 (*p* < 0.001) and from week 2 to week 4 (*p* < 0.001), which corresponded with the natural antiviral response to the two vaccinations for all the antigens tested (Figure 2a,b). Neutralizing antibodies were maintained at values up to 80% during the next 10 weeks of follow-up (Figure 2a). The titer of IgG antibodies against the SARS-CoV-2 spike, S1, and trimer proteins showed similar significant patterns, with differences in their relative abundance. At week 6, antibodies against the S1 region exhibited the highest titer (Appendix A) followed by that against the trimer region (Appendix A), while those against the RBD presented the lowest levels (Figure 2b and Appendix A). Plasma levels of the RBD protein was measured as it is present in both vaccines in the full-length spike protein and is temperature-stable [18]. Given the inter-individual differences in the in vitro capacity to neutralize the virus observed in our cohort 10 weeks after vaccination, we were able to split the cohort into LN and HN groups as described above to examine immunologic and gut microbial differences between them. The pattern of IgG antibodies against the SARS-CoV-2 spike–RBD region exhibited a distribution pattern of high and low titer when plotted as LN and HN groups, in which neutralization capacity was significantly lower in the LN group at week 8 and 10 (*p* < 0.05) (Figure 2c). This trend was similarly observed when the trajectory of RBD antibodies was re-evaluated by group, suggesting the LN group may have experienced an attenuated response to the vaccine (Figure 2d).

### 3.2. Inflammation Status and Inflammatory Responses

An exploratory panel of 65 inflammatory biomarkers was measured in plasma samples using Luminex technology to assess inflammation status pre-vaccination and stratified by neutralization capacity group. At the pre-vaccination timepoint, there were distinct inflammatory profiles for both neutralization groups. In general, inflammatory cytokines were more abundant in the LN group compared to the HN group (Figure 3a). The LN group showed higher levels of MIF (*p* < 0.001), MIP-1alpha (*p* < 0.05), MCP-1 (*p* < 0.05), MIP-1beta (*p* < 0.01), Eotaxin (*p* < 0.01), IL-17 (*p* < 0.001), MCP-2 (*p* < 0.01), CD30 (*p* < 0.01), VEGF-A (*p* < 0.05), and IP-10 (*p* < 0.001) compared to the HN group. Conversely, the HN group generally exhibited lower levels of inflammatory biomarkers but showed higher plasma concentrations of SCF (*p* < 0.05), MDC (*p* < 0.001), and MIP-3-alpha (*p* < 0.01) when compared to the LN group (Figure 3b). This suggests that individuals in the LN group likely experienced chronic inflammation prior to vaccination, which potentially interfered with the vaccine response and may explain their lower degree of viral neutralization.

### 3.3. Gut Microbiome Composition

We next examined the gut microbiome of participants to investigate its potential role in viral neutralization capacity. Temporal changes in the abundance of gut microbiota pre- and post-vaccination revealed distinct patterns associated with viral neutralization (Appendix A). Compared to their pre-vaccination levels of the cohort overall, we observed a decreased abundance in the following taxa 10 weeks post-vaccination: *Ruminococcus torques*, *Anaerotruncus colihominis*, *Butyrivibrio*, *B. crossotus*, *Prevotella*, *P. copri*, and *Eubacterium* (Figure 4a). Generally, gut microbe abundance decreased following vaccination (Figure 4b). When assessing the impact of vaccination on the gut microbiome based on eventual viral neutralization capacity, we observed distinct compositional patterns (Figure 4c). Specifically, in the LN group, *Eubacterium*, *Butyrivibirio*, *B. crossotus*, *Prevotella*, *P. copri*, *Anaerotruncus colihominis*, and *Ruminococcus torques* were more abundant compared to the HN group pre-vaccination. However, by week 10 post-vaccination, we observed significantly lower abundances of these taxa in both groups (Figure 4c). Comparatively, *Bacteroides stercorirosoris*, *B. massiliensis*, *Bifidobacterium bifidum*, *Coprococcus catus*, and *Holdemania filiformis* sp. were present at high abundance in the HN group pre-vaccination and decreased at week 10 post-vaccination. Species such as *Alistipes putredinis*, *Coprococcus comes*, *Bacteroides salyersiae*, *B. massiliensis*, and *Sutterella stercoricanis* were detected at moderate levels, gradually decreasing by the final week of sampling (Figure 4c). Greater abundance of butyrate-producing bacteria, including *Coprococcus comes*, *Alistipes putredinis*, *Eubacterium ramulus*, and *Roseburia* sp., (more abundant in the HN group), were observed. These data suggest that the microbes exhibiting differential abundance between the neutralization groups pre-vaccination may partly contribute to the observed inter-individual differences in vaccine response. Additionally, vaccination itself may lead to a likely transient reduction in the abundance of gut microbes. Differentially abundant taxa observed pre- and post-(10 week) vaccination are listed in Appendix A.

### 3.4. Inflammation Pathway Analysis

The Ingenuity Pathway Analysis (IPA, v01-22-01, QIAGEN Inc., https://digitalinsights.qiagen.com/IPA, accessed on 21 May 2024) software [19] was utilized to identify enriched pathways in our cohort using the 65-plex exploratory panel of inflammatory biomarkers at the pre-vaccination timepoint. IPA highlighted several pathways that involve immune responses commonly implicated in vaccine response or disease (Figure 5a). Among these enriched pathways, the HMGB-1 signaling pathway was of particular interest, given its central role in inflammation and association with inflammatory responses (Figure 5b). Longitudinal HMGB-1 profiling showed higher concentrations pre-vaccination in the LN group compared to the HN group. However, following vaccination, HMGB-1 levels became similar between the groups over time (Figure 5c). To further examine the relationship between pre-vaccination HMGB-1 levels and immune response following vaccination, we examined the association between plasma levels of HMGB-1 and neutralizing antibody capacity at 6, 8, and 10 weeks post-vaccination (Figure 5d). Consistently, significant negative correlations were observed between pre-vaccination HMGB-1 levels and neutralization capacity across all three timepoints from week 6 through 10 (Pearson = −0.47, −0.62, and −0.54). These findings suggest that pre-vaccination HMGB-1 levels may serve as a potential biomarker of vaccine effectiveness measured by in vitro neutralization. Given its known role in immune cell propagation, NO was measured to further examine possible roles of HMGB-1 in vaccine response. The Pearson correlation analysis revealed that NO levels pre-vaccination positively correlated with HMGB-1 levels measured at pre-vaccination and week 10 post-vaccination timepoints (0.46 and 0.52, respectively) (Figure 5d).

### 3.5. Immune Response and Gut Microbiota Composition

We conducted a multivariate correlation analysis to investigate the relationship between the pre-vaccination levels of HMGB-1, inflammation biomarkers of interest, and gut microbial taxa that previously exhibited differences in abundance between pre- and post-(10-week) vaccination timepoints as described in Figure 4a. Significant correlations were observed between different immunological biomarkers and gut microbiota within the overall cohort (Figure 6). Bacterial taxa such as *Anaerotruncus colihominis*, *Butyrivibiro crossotus*, *Prevotella copri*, and the genera *Butyrivibrio* and *Prevotella* were positively correlated with HMGB-1, while *Bacteroides faecis*, *Bifidobacterium bifidum*, *Collinsella intestinalis*, *Megamonas funiformis*, and *Ruminococcus gnavus* correlated negatively. Interestingly, *Butyrivibiro crossotus*, *Prevotella copri*, and the genera *Butyrivibrio* and *Prevotella* that were positively correlated with HMGB-1 concentrations were conversely negatively correlated to most of the inflammatory biomarkers evaluated. These bacterial taxa were also found at a higher abundance in the LN group which further indicated the significance of HMGB-1 in predicting lower neutralization capacity (Figure 4c). Bacterial taxa that were found at higher abundance in the HN group such as *Bifidobacterium bifidum*, *Collinsella intestinalis*, *Megamonas funiformis*, and *Ruminococcus gnavus* were found to be positively correlated with the same panel of inflammatory biomarkers (Figure 4c and Figure 6).

## 4. Discussion

To study the inter-individual differences in vaccine response to SARS-CoV-2, we investigated pre-vaccination states of the gut microbiome and inflammation, as well as changes to these factors following vaccination. We were able to discern such differences by stratifying our pilot cohort of vaccinees by in vitro neutralization capacity over a 10 week period while tracking vaccine-related side effects, as described in Figure 1. Covariates associated with differences in vaccine response including ethnicity, sex, age, and BMI did not appear to be significant factors that contributed to inter-individual differences in viral neutralization capacity following vaccination in our small cohort. However, other factors such as sociocultural or lifestyle differences that may influence vaccine response and gut microbiome, such as education diet, could not be accounted for as this pilot study was not sufficiently powered for this assessment given our low sample size.

Despite this limitation, we observed significant differences in the inflammatory state and gut microbiome composition that associated with inter-individual differences in immune response to the vaccine. Indeed, the levels of antibodies produced against SARS-CoV-2 antigens significantly increased over time post-vaccination (Figure 2b and Appendix A). This demonstrated that the vaccine produced a sufficient immune response, however significant inter-individual differences in neutralization capacity were noted. At 10 weeks post-vaccination, anti-SARS-CoV-2 antibody production titers showed the greatest variability among the cohort; antibodies against the S1 and spike trimer antigens showed the largest differences, and those against the RBD antigen trended downward by week eight. Pre-vaccination cross-reactivity was on average 10.0% (±7) with no significant differences between high and low neutralization groups which could be due to cross-reactive antibodies present from previous coronavirus infections [20]. Although specific response is lost over time, the neutralizing antibodies were maintained throughout study period. Additionally, previous studies have shown that prior SARS-CoV-2 exposure can enhance the production of neutralizing antibodies induced by vaccination, which may impact the effectiveness of the vaccine and immunity against emerging viral variants [21]; rapid mutations in the viral genome can also affect the host’s neutralization capacity [22].

Chronic inflammation involves cytokine dysregulation and plays a significant role in the development of various pathologies [23]. We examined the relationship between pre-vaccination inflammation states and vaccine-induced immunity using Luminex assays. Our results suggest a pattern of inflammatory biomarkers or immune signature pre-vaccination that associates with later neutralization capacity. In particular, the LN group displayed elevated levels of MIF, MCP1, MIP-1alpha, MIP-1beta, and IP-10, whereas the HN group had lower levels of these inflammatory biomarkers pre-vaccination. These biomarkers contribute to various aspects of the immune response, including inflammation, immune cell recruitment, and regulation of immune cell function, and they play important roles in the response to viral infections. However, their presence prior to vaccination may indicate immune activation by other infections or chronic inflammation states that may attenuate antibody production against SAR-CoV-2 upon vaccination.

Elevated levels of these inflammatory biomarkers were observed in COVID-19 patients and play a possible role in contributing to COVID-19 severity [14]. In particular, the cytokines MCP1, MIP-1alpha, MIP-1beta, and IP-10 are often overexpressed in patients that experience more severe COVID-19 symptoms in what is characterized as “cytokine storms” [23,24]; we observed significantly more abundant plasma levels of these cytokines in the LN group compared to the HN group (Figure 3). Interestingly, HMGB-1 interacts with these cytokines, as well as the ligands and receptors associated with elevating pro-inflammatory cytokines, resulting in cytokine storms [25]. The high levels of HMGB-1 pre-vaccination among individuals in our cohort coupled with the higher levels of these downstream cytokine targets of HMGB-1 may explain the lower vaccine-induced neutralization capacity observed. This is consistent with the positive correlation between HMGB-1 and NO, which may be responsible in attenuating a B-cell response by regulating BAFF.

HMGB-1 levels may be influenced by the gut microbiome, which is known to regulation inflammation and immune response to viral infections. In addition, microbial metabolites like SCFAs impact health generally, with implications to vaccine effectiveness [5,6]. SCFAs produced by the gut microbiome include butyrate, acetate, and propionate which support the energy needs in maintaining B-cells through increasing glycolysis, fatty acid synthesis, and oxidation [26]. Butyrate was also associated with the inhibition of HMGB-1 [27]. Butyrate can also influence the intestinal barrier through upregulation of mucin 2 and the expression of tight junction proteins which decrease permeability [28]. While butyrate at lower concentrations can promote intestinal barrier integrity, butyrate present at higher concentrations is implicated in inducing apoptosis [29]. In our study, we observed that microbiome taxa such as *Eubacterium ramulus* and *Roseburia* spp. were present at a higher abundance in the HN versus the LN group, and were positively correlated with butyrate production, which altogether could further influence HMGB-1 levels [30,31,32]. In contrast, microbes in the genus *Prevotella* were found at a higher abundance in the LN versus the HN group. *Prevotella* stimulates TH-17 through the activation of TLR-2 and release of cytokines such as IL-1 and IL-23 and is associated with increases in inflammation through stimulating epithelial cells to further increase cytokine production [33]. Higher inflammation was associated with the release of HMGB-1 due to increased cellular stress [27]. The interaction of *Prevotella* with these pro-inflammatory pathways can increase the risk of chronic inflammation and gut dysbiosis [32].

Furthermore, the natural adjuvant hypothesis theorizes that adjuvants from microbiota could enhance or hinder the immune response of the host [4]. Adjuvants have been used in vaccines to enhance vaccine response [34]. However, it is possible that natural adjuvants produced within the body by microbiota could also have a similar effect. Prior research showed that adjuvants produced by gut microbiota could be influenced by gut epithelial integrity [4]. Lower gut epithelial integrity could increase the adjuvants available to the body. However, gut leakage is also associated with less desirable health outcomes in the host. In this way, microbiota can play a role in creating individual differences in the immune response to the vaccine.

HMGB-1 can be released extracellularly through multiple processes. Apoptosis, pyroptosis, and NETosis are forms of cell death that can result in HMGB-1 release [35]. TLR-4 activation is one way that HMGB-1 is released [36]. Lipopolysaccharides (LPSs) are gut-derived mediators that can activate TLR-4 and is largely produced by the gut microbiota [37]. While LPSs are mostly processed in the liver, overgrowth of bacteria in the gut or lowered gut barrier integrity can allow increased LPS to be present in the body [34,35]. Stimulation of the LPS-TLR4 signaling pathway can result in the release of HMGB-1 and it can also cause the release of other proinflammatory factors such as TNFα and IL6 [25]. Gram-negative bacteria such as *Prevotella* which was found in higher abundance in the LN group are known to produce LPSs that interact with TLR-4 which could also elevate HMGB-1 levels [37,38,39]. Furthermore, within this cohort *Anaerotruncus colihominis*, *Butyrivibrio crossotus*, and *Provotella copri* positively correlated with HMGB1 and were found in higher abundance pre-vaccination in the LN group, while *Eubacterium*, *Ruminococcus*, and *Collinsella instestinalis* were negatively correlated with HMGB-1 and present in higher abundance in the HN group (Figure 3 and Figure 6).

HMGB-1 has been implicated in several studies demonstrating its utility as both a biomarker of COVID-19 prognosis and risk factor of developing long COVID symptoms [14,40,41]. This study further adds to this body of knowledge by highlighting HMGB-1 as a promising biomarker that may assist in predicting SARS-CoV-2 vaccine outcomes. Together, our results indicate an immune-microbial signature pre-vaccination that associates with vaccine-induced neutralization capacity, with implications beyond COVID-19 vaccines. However, given our limited sample size, a study of a larger, more diverse cohort of vaccinees, is needed to validate our initial findings and extend their generalizability. Other studies which consider the origin and isoform of HMGB-1 should also be considered as we did not explore this in our study. Our findings thus warrant further investigation into the understanding the relationship between HMGB-1 and immune response, as well as its interaction with intestinal microbial taxa, in shaping vaccination effectiveness and potentially reducing related side effects to enable a more personalized approach to vaccination. 

## Figures and Tables

**Figure 1 vaccines-12-00637-f001:**
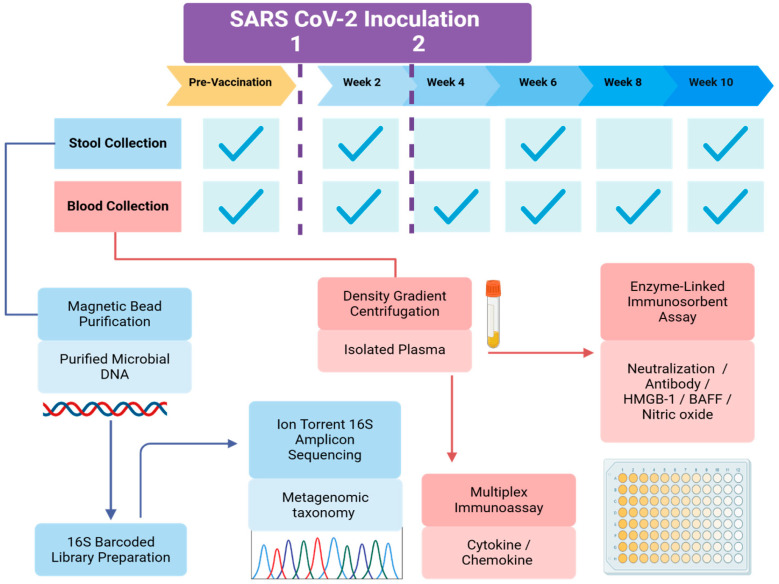
General study design. Flowchart depicting this longitudinal observational pilot study to characterize inflammatory and gut microbiome changes in response to SARS-CoV-2 vaccination. Blue lines/arrows represent the flow of processing stool samples. Red lines/arrows indicate the process of isolating and assaying plasma from blood samples. The dotted purple lines indicate approximate timing of the 1st and 2nd doses of the SARS-CoV-2 vaccine within the study timeline.

**Figure 2 vaccines-12-00637-f002:**
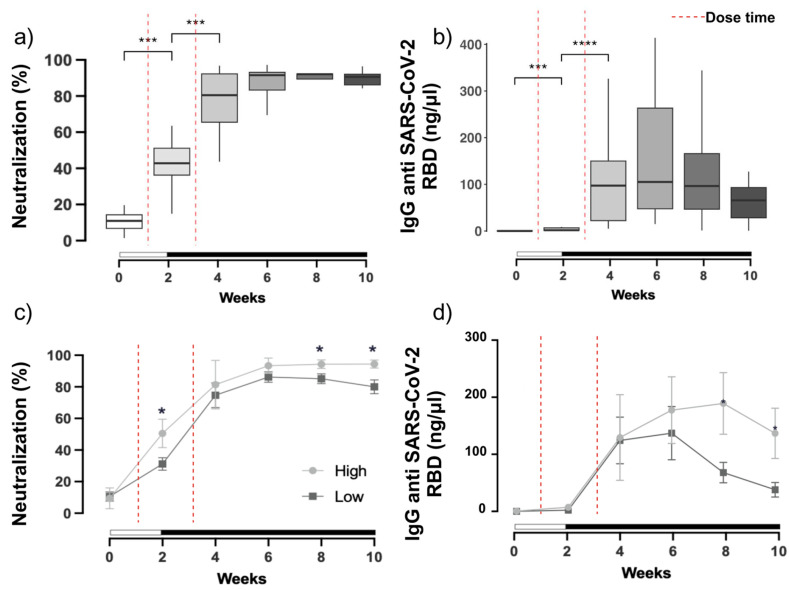
Temporal dynamics of SARS-CoV-2 immune response in the cohort overall and stratified by neutralization group indicated. (**a**) Anti-viral neutralization capacity (% inhibition) compared stepwise across two-week intervals. (**b**) Concentrations (ng/µL) of RBD IgG antibodies compared stepwise across two-week intervals. (**c**) Inter-group significance determined via Dunn’s *post hoc* test. Anti-viral neutralization capacity divided into high or low neutralization groups based on the tenth week post-vaccination timepoint. (**d**) Concentrations (ng/µL) of RBD IgG antibodies stratified by high or low neutralization group. * *p* < 0.05; *** *p* < 0.001; and **** *p* < 0.0001. The dashed red line indicates the time of the first and second dose of the vaccine.

**Figure 3 vaccines-12-00637-f003:**
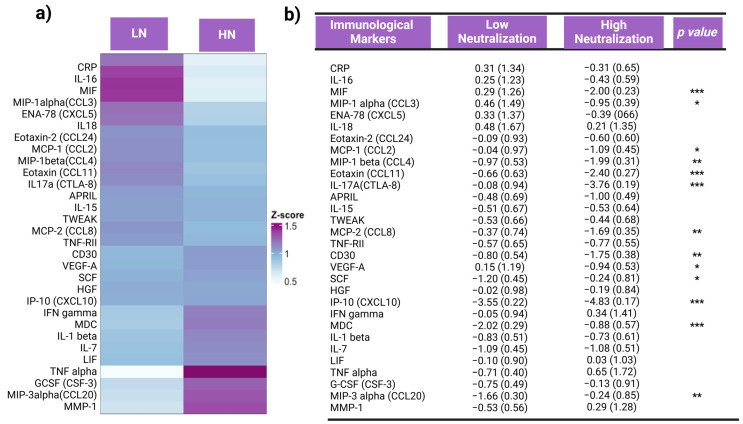
Differences in the relative levels of inflammatory biomarkers stratified by neutralization group. (**a**) Hierarchical heatmap of mean normalized values. (**b**) Unpaired *t*-test of the z-transformed data, displaying mean, standard deviation, and significance (* *p* < 0.05; ** *p* < 0.01; and *** *p* < 0.001). Inflammatory biomarkers exhibited an inverse relationship between low and high neutralization groups, with the LN group demonstrating a higher mean normalized abundance of inflammatory biomarkers in general.

**Figure 4 vaccines-12-00637-f004:**
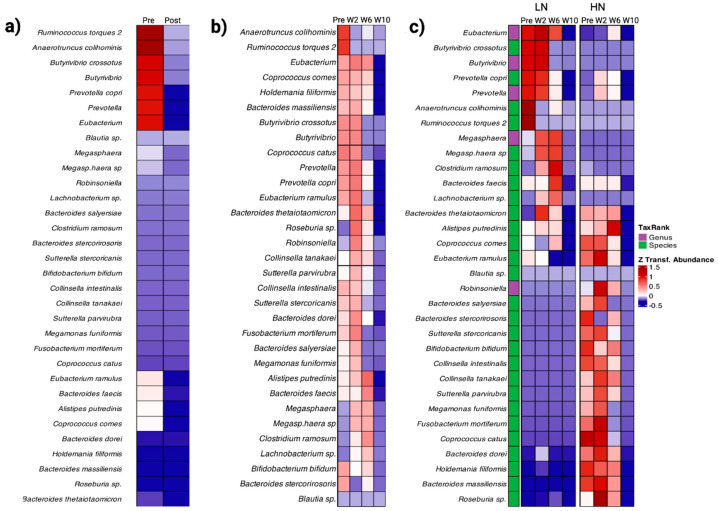
Heatmaps showing the temporal changes in gut microbiota composition exhibiting significant differences in mean relative abundance following vaccination. (**a**) Mean abundance of the indicated taxa that were identified to be significant pre- and post-vaccination measured at 10 weeks across all individuals. (**b**) Changes in the abundance of indicated microbial taxa in panel **a** following vaccination across all individuals per timepoint shown. (**c**) Changes in the relative abundance between the indicated taxa as in panel **a** stratified by LN and HN groups pre-vaccination and 2-, 6-, and 10-weeks post-vaccination. Bacterial composition differed between the two groups, showing inverse abundance of taxa. Results are given as a z-transformed abundance score for all heatmaps. Pre: pre-vaccination; Post: post-vaccination; W2,6,10: 2, 2-, 6-, and 10-weeks post-vaccination.

**Figure 5 vaccines-12-00637-f005:**
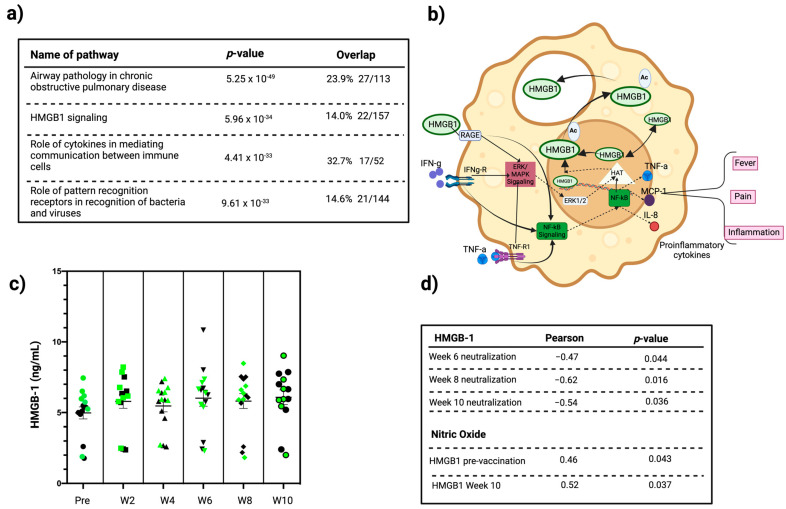
Examining plasma levels of HMGB-1 in response to SARS-CoV-2 vaccination and associations with neutralization capacity and plasma NO. (**a**) IPA using the 65-plex Human ProcartaPlex Panel for inflammation biomarkers. (**b**) HMGB-1 signaling pathway representation in immune cells. (**c**) Temporal changes in HMGB-1 concentrations at the indicated timepoints pre-vaccination (Pre), and 2–10 weeks post-vaccination (W2-W10), with green and black dots denoting the LN and HN group, respectively. (**d**) Pearson correlation analysis of pre-vaccination HMGB-1 levels with neutralizing antibody capacity at 6-, 8-, and 10-weeks post-vaccination, and the relationship between plasma levels of HMGB-1 with nitric oxide pre-vaccination and at 10 weeks post-vaccination.

**Figure 6 vaccines-12-00637-f006:**
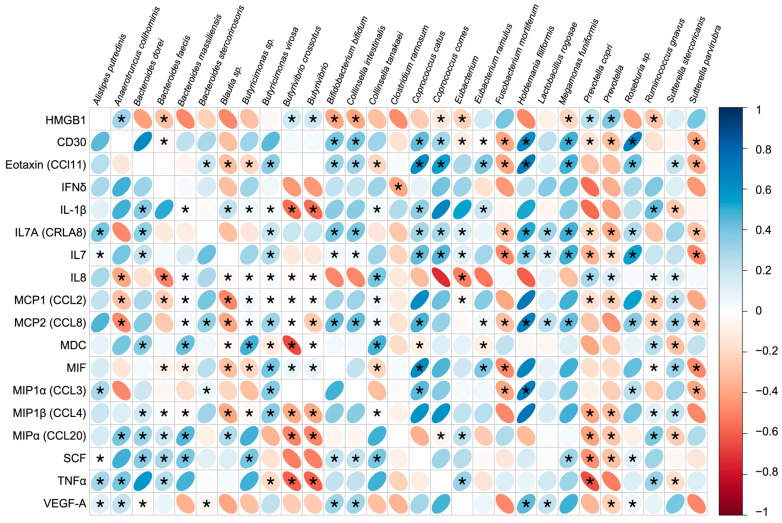
Correlation matrix of inflammatory biomarkers in relation to the microbiota taxa that exhibited significant differences between pre-vaccination and post-(10 week) vaccination timepoints. Comparisons between the plasma levels of these biomarkers and the mean abundance of indicated bacterial taxa of the overall cohort were at the pre-vaccination timepoint. The Spearman’s rank correlation coefficients were created with the corrplot Bioconductor package (v0.92); * *p* < 0.05.

**Table 1 vaccines-12-00637-t001:** Biometric and demographic characteristics of the participants by low (LN) or high (HN) neutralization group.

Characteristics		Total	LN (n = 7)	HN (n = 7)	*p*-Value
Age (±SD), age		32.1 (±9.9)	32.7 (±12.2)	31.4 (±8.2)	0.820
Residence time in Hawaii, years (±SD)		14.1 (±10.3)	14.2 (±10.4)	14.0 (±10.9)	0.981
Baseline cross-reactivity, pre-vaccine, % (±SD)		10.0 (±7.0)	10.6 (±7.9)	9.5 (±6.6)	0.768
BMI	kg/m^2^ (±SD)	25.3 (±4.5)	24.4 (±6.0)	26.2 (±2.4)	0.472
Sex	Female	7 (50%)	4 (57%)	3 (43%)	>0.999
	Male	7 (50%)	3 (43%)	4 (57%)	
Ethnicity	Asian	8 (57%)	4 (57%)	4 (57%)	>0.999
	Non-Asian	6 (43%)	3 (43%)	3 (43%)	
Vaccine brand	Pfizer	11 (50%)	6 (86%)	5 (72%)	0.768
	Moderna	3 (50%)	1 (14%)	2 (28%)	

## Data Availability

All data used for this project will be available de-identified when approved by the University of Hawaii Institutional Review Board upon reasonable request to the corresponding author. The gut microbiome data presented in the study are deposited in the figshare repository, accessible at: https://doi.org/10.6084/m9.figshare.25892719.v1, accessed on 21 May 2024.

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
