# Peer review of "Immuno-Microbial Signature of Vaccine-Induced Immunity against SARS-CoV-2"

_vaccines, 2024, doi:10.3390/vaccines12060637_

Round 1

Reviewer 1 Report

Comments and Suggestions for Authors

The manuscript is very interesting and has good practice. It shows gut microbiota may influence neutralizing antibody levels and inflammation response after SARS-Cov-2 mRNA inoculation. However, the major issues are: (1) the same size, this study is smaller same size and the results need to be confirmed. For example, the Pearson correlation (R) for HMGB-1 is only -0.5 at week 10, it is not a good linear correlation. (2) the sample size is 14 in total, LN group and HN group are the same as 7. Does it have sample selection bias? The minor issues are: (1) the authors should explain more detail about LN and HN groups. It would be easy to understand LN and HN groups if use a table for supplementary figure A1. (2) the authors should describe the results in more detail with mean±SD in text or have tables, not just figures. (3) In the discussion section, it would be better if the authors discussed more about the possible mechanism for LN group with high inflammation response and the relationship with the risk of cytokine storm.

Comments on the Quality of English Language

Author Response

We thank the reviewer for the many positive and constructive comments to our manuscript. We feel that our response to these comments greatly improved the manuscript, for which we are grateful. Please see the following point-by-point response, which we hope you will find satisfactory.

Comments and Suggestions for Authors

The manuscript is very interesting and has good practice. It shows gut microbiota may influence neutralizing antibody levels and inflammation response after SARS-Cov-2 mRNA inoculation. However,

the major issues are:

 (1) the same size, this study is smaller same size and the results need to be confirmed. For example, the Pearson correlation (R) for HMGB-1 is only -0.5 at week 10, it is not a good linear correlation.

We acknowledge your observation regarding the low correlation. However, it is important to clarify that the primary objective of this study is to assess the statistically significant impact of HMGB-1 levels on neutralization, rather than focusing on predicting or determining the magnitude of the variable of interest. We will make our objectives clearer in the text and describe that this may be a limitation of our results.

(2) the sample size is 14 in total, LN group and HN group are the same as 7. Does it have sample selection bias?

We employed the 50th percentile as a cut-off threshold. Consequently, it encompasses both ends of the spectrum equally, with no samples excluded from the analysis.

The minor issues are:

 (1) the authors should explain more detail about LN and HN groups. It would be easy to understand LN and HN groups if use a table for supplementary figure A1.

Thank you for your suggestion. We agree that having a table describing the low and high neutralization groups in more detail would be beneficial to readers. We have added Table 1 that better describes LN and HN groups in the main text, not as a supplementary.

Table 1: Biometric and demographic characteristics of participants categorized into the viral neutralization groups: High Neutralization (HN) or Low Neutralization (LN) group.

Characteristics

Total

LN (n=7)

HN (n=7)

p-value

Age, year (±SD)

32.1 (±9.9)

32.7 (±12.2)

31.4 (±8.2)

0.820

Residence time in HawaiÊ»i, years (±Sd)

14.1 (±10.3)

14.2(±10.4)

14.0(±10.9)

0.981

Inhibition assay average, before vaccination, percent (±SD)

10.0 (±SD)

10.6(±7.9)

9.5(±6.6)

0.768

BMI

(Kg/m2), (±SD)

25.3(±4.5)

24.4(±6.0)

26.2(±2.4)

0.472

Gender

Female

7(50%)

4(57%)

3(43%)

>0.999

Male

7(50%)

3(43%)

4(57%)

Ethnicity

Asian

8(57%)

4(57%)

4(57%)

>0.999

Non-Asian

6(43%)

3(43%)

3(43%)

Vaccine Brand

Pfizer

11(50%)

6(86%)

5(72%)

0.768

Moderna

3(50%)

1(14%)

2(28%)

(2) the authors should describe the results in more detail with mean±SD in text or have tables, not just figures.

Thank you for this suggestion. In the new version of the manuscript, we provide a more detailed report of participant demographics, grouped by LN and HN. The table includes both mean and SD.

(3) In the discussion section, it would be better if the authors discussed more about the possible mechanism for LN group with high inflammation response and the relationship with the risk of cytokine storm.

We appreciate your suggestions, and we agree that further describing potential mechanisms involved in the inflammatory response regarding cytokine storms would be beneficial. We have added a section to describe potential pathways that may influence this response.

Reviewer 2 Report

Comments and Suggestions for Authors

Comments and Suggestions:

The article “Immuno-microbial signature of vaccine-induced immunity of SARS-CoV-2” by Umeda et. al., is very well written and would have been helpful if published a couple of years ago. The manuscript provides an insight of HMGB-1 as a potential indicator for predicting vaccine effectiveness in healthy individuals within the population in Hawai’i.  

Clarity and Structure: The introduction effectively sets the stage by presenting thorough explanation about COVID-19 history, immune responses, vaccination and gut microbiota.

I do have some comments:

1.      There are already many articles claiming HMGB1 as SARS-CoV2 infection biomarker, such as PMID: 33313438, PMID: 36631939, PMID: 38504105 and more. I am concerned with the lack of novelty in the submitted manuscript. Please explain about your novel findings compare with these previous reported studies.

2.      The authors are suggested to add detailed demographic, diagnostics, and biochemical characteristics of study subjects used in this study. What is their medical history in terms of comorbidities such as TB or HIV or other lung infections?

3.      Please mention the IRB or ethical committee Protocol approval Number, if any.

4.      If the samples collected for this study are so old (from Jan to April 2021), then why the delay in manuscript? Have authors considered other COVID-19 waves and changes in the findings with respect to new strains of SARS-CoV2? Were these subjects tested for SARS-CoV2 infections? What tests were done?

5.      The sample size was very less (n=14) to reach to a conclusion that HMGB-1 is the potential biomarker of in predicting SARS-CoV-2 mRNA vaccine effectiveness. Please mention the limitations of this study.

Author Response

We thank the reviewer for your constructive critiques and provide a point-by-point response here. We feel that your comments helped us to improve our manuscript in the new version we resubmit for consideration and are grateful for your insights.

The article “Immuno-microbial signature of vaccine-induced immunity of SARS-CoV-2” by Umeda et. al., is very well written and would have been helpful if published a couple of years ago. The manuscript provides an insight of HMGB-1 as a potential indicator for predicting vaccine effectiveness in healthy individuals within the population in Hawai’i. 

Clarity and Structure: The introduction effectively sets the stage by presenting thorough explanation about COVID-19 history, immune responses, vaccination and gut microbiota.

I do have some comments:

  1. There are already many articles claiming HMGB1 as SARS-CoV2 infection biomarker, such as PMID: 33313438, PMID: 36631939, PMID: 38504105 and more. I am concerned with the lack of novelty in the submitted manuscript. Please explain about your novel findings compare with these previous reported studies.

Thank you for bringing attention to the opportunity to elucidate the novelty of our study findings. Although the studies mentioned in your comment illustrate the role of HMGB1 as a biomarker for COVID-19 prognosis, none specifically investigate its utility in assessing vaccine response. Moreover, our study uniquely focuses on pre-existing inflammatory profiles and gut microbiome composition in relation to vaccine effectiveness. Given the linkage of HMGB1 to these systems, its prominence in our data makes it a compelling target for further investigation when considering vaccine effectiveness. We have made this distinction about the novelty of our research clearer in the manuscript.

The main distinction from previous studies and the novelty in this study lies in its focus on a median productive age healthy population. Our initial findings aim to provide valuable insights for predicting potential outcomes in healthy individuals, while also assisting stakeholders in making informed decisions by considering the diverse characteristics of the population.

It's important to acknowledge that there are still two major challenges associated with these designed vaccines. Firstly, there's a concern regarding the low mucosal response, particularly in the upper respiratory system. Secondly, the fast replication of the virus presents an ongoing issue. Multivalent vaccines have demonstrated better immunological responses and offer some protection against new viral variants. However, it's worth noting that none of the current biological products provide complete disease prevention. Additionally, there's a need to consider individual differences in pre-existing inflammatory profiles and gut microbiome composition when designing vaccines, which makes our study a highly novel.

  1. The authors are suggested to add detailed demographic, diagnostics, and biochemical characteristics of study subjects used in this study. What is their medical history in terms of comorbidities such as TB or HIV or other lung infections?

We appreciate the reviewers’ suggestions and agree that a more detailed demographics table would be beneficial. We have modified Table 1 that is more informative of the study subjects. Regarding medical history concerning TB and HIV, we can confirm study participants were not diagnosed with either at the time this study was conducted. This study was conducted in a healthy cohort from a convenience sampling.

  1. Please mention the IRB or ethical committee Protocol approval Number, if any.

We confirm that the protocol for our study was approved by the University of Hawaiʻi Office of Research Compliance under Protocol ID 2019-00376, approval date May 1, 2021, and all procedures were conducted in strict adherence to the guidelines outlined therein. Informed consent was obtained from all participants, prior to their involvement in the study. We have ensured that these details are appropriately included in the manuscript. This information is now located between lines 110-113 of the new version of the manuscript. Additionally, it is also included in the corresponding section: Institutional Review Board Statement.

  1. If the samples collected for this study are so old (from Jan to April 2021), then why the delay in manuscript? Have authors considered other COVID-19 waves and changes in the findings with respect to new strains of SARS-CoV2? Were these subjects tested for SARS-CoV2 infections? What tests were done?

Thank you for clarifying. The objective of your study is to enhance understanding of the individual conditions within the population that might hinder the response to any vaccine. Vaccines against SARS-CoV-2 still need refinements in their design due to the noted low immunological response in the upper respiratory tract and the rapid rate of virus mutations. Therefore, the goal of limiting infection remains an ongoing challenge, including the need for approaches that promote personalized medicine. We also note that subjects in the study were tested for SARS-CoV-2 and these results were part of our eligibility criteria for enrollment. We clarify this in the reviewed manuscript.

It's important to note that our study does not differentiate between virus strains; rather, it focuses on evaluating vaccine effectiveness based on a generalized response implicating the connections between inflammation, the gut microbiome, and HMGB-1.

We also understand that there was a delay in between us obtaining the samples and submitting the article. Because of constraints in our funding, it took us a certain amount of time to finish the analysis and write the article. In addition, it was very challenging to remain active during pandemic as our institution faced many barriers and restrictions, which contributed to this delay. We insist, however, that the discoveries of this article are still relevant and important to the community.

  1. The sample size was very less (n=14) to reach to a conclusion that HMGB-1 is the potential biomarker of in predicting SARS-CoV-2 mRNA vaccine effectiveness. Please mention the limitations of this study.

We appreciate the reviewer's consideration of the sample size in our pilot study. Indeed, the sample size of 14 participants is relatively small, and we acknowledge this as a limitation of our study. We have included a discussion of these points in the revised manuscript that address given the limited sample size, we recognize that our findings may not be conclusive in establishing HMGB-1 as a definitive biomarker for predicting SARS-CoV-2 mRNA vaccine effectiveness. We acknowledge that larger studies with more participants are needed to validate our preliminary findings and draw more robust conclusions and mention that our study was conducted within a specific population or setting, and the generalizability of our results may be limited.

Reviewer 3 Report

Comments and Suggestions for Authors

The paper shows remarkable results with small samples of diverse patients. There is also a fine demarcation line between the neutralising titres of the high and low responders although the IgG anti-RBD antibodies look more substantial with respect to the persistence of the high titres. Perhaps this warrants more comment.

Figure 3 for the immune markers requires explanation. Are the normalised data in Figure 3a based on the deviation from the combined HN and LN calculation ? Are the mean values deviations from the combined mean? Why is the TNF alpha box blank in the LN group in Figure 3a and why does the text refer to the reduced value of TNF alpha in the LN when it wasn't significant?

The paper does not attempt to link the very interesting HMGB1 data with the HMGB1 findings in reference 4. It would be obvious to  look for associations with diet, ethnicity and BMI but that is waiting for a larger sample.

Author Response

Reviewer 3

We thank the reviewer for the many helpful comments and recommendations that improved our manuscript. In the following point-by-point response, we hope you find our modifications to the manuscript satisfactory.

Comments and Suggestions for Authors

  1. The paper shows remarkable results with small samples of diverse patients. There is also a fine demarcation line between the neutralising titres of the high and low responders although the IgG anti-RBD antibodies look more substantial with respect to the persistence of the high titres. Perhaps this warrants more comment.

Thank you for your insightful comment. We acknowledge the importance of the distinction between neutralizing antibody titers among high and low responders. To address this, we included additional commentary in the discussion section to elaborate on the observation regarding the more substantial presence of IgG anti-RBD antibodies in individuals with high titers. We discuss the implications of this finding for vaccine effectiveness and the persistence of immune responses over time. We believe that this additional discussion enriches the interpretation of our results and offer valuable insights into the mechanisms underlying vaccine-induced immunity.

  1. Figure 3 for the immune markers requires explanation. Are the normalised data in Figure 3a based on the deviation from the combined HN and LN calculation ?

Thank you for your inquiry. Following the completion of luminex assays, it became apparent that the inflammation factors exhibited variations across different scales. To facilitate clearer visualization of the disparities between low and high neutralization groups, we employed a Z-Score Normalization. This transformation ensured that the data were standardized with a mean of 0 and a standard deviation of 1.5, facilitating robust comparisons across the various immune markers. We believe this approach enhances the clarity and comprehensibility of Figure 3a in illustrating the observed trends in immune marker expression.

  1. Are the mean values deviations from the combined mean?

Thank you for your question. As said above, yes, the values on 3b are data standardized with a mean of 0 and a standard deviation of 1.5, facilitating robust comparisons across the various immune markers.

  1. Why is the TNF alpha box blank in the LN group in Figure 3a and

We appreciate the reviewer's attention to detail regarding the TNF-α "box" in Figure 3a, where it is depicted with the lowest z-score for the LN group, resulting in a white appearance. To address this concern and enhance clarity, we have enlarged the legend to provide better visibility and understanding of the heatmap. Thank you for highlighting this issue, and we trust that these adjustments will improve the readability of our manuscript.

  1. why does the text refer to the reduced value of TNF alpha in the LN when it wasn't significant?

We appreciate the reviewer's feedback regarding the inclusion of TNF-α in our manuscript. Upon careful consideration, we recognize the potential confusion this may have caused. Initially, our rationale for including TNF-α was to underscore its significant connection to both the immune system and the gut microbiome (presented later in the data). However, in response to the reviewer's suggestion, we have removed the mention of TNF-α in the result description for Figure 3a. Thank you for bringing this to our attention, and we believe this adjustment will enhance the clarity and focus of our work.

  1. The paper does not attempt to link the very interesting HMGB1 data with the HMGB1 findings in reference 4. It would be obvious to look for associations with diet, ethnicity and BMI but that is waiting for a larger sample.

Thank you for your thoughtful comment. Unfortunately, as stated in the article we did not survey diet at the time of the study. Given the limited sample size, we could not assess differences by BMI or ethnicity. However, we hope findings inspire further studies in a larger sample to examine these very important questions.

Round 2

Reviewer 1 Report

Comments and Suggestions for Authors

The authors answered all my questions. I have no other questions for this manuscript.